# Estimated effectiveness of symptom and risk screening to prevent the spread of COVID-19

Katelyn Gostic[1]*, Ana CR Gomez[2], Riley O Mummah[2], Adam J Kucharski[3], James O Lloyd-Smith[2,4]*

[1]Department of Ecology and Evolution, University of Chicago, Chicago, United States; [2]Department of Ecology and Evolutionary Biology, University of California, Los Angeles, Los Angeles, United States; [3]Department of Infectious Disease Epidemiology, London School of Tropical Hygiene and Medicine, London, United Kingdom; [4]Fogarty International Center, National Institutes of Health, Bethesda, United States

**Abstract** Traveller screening is being used to limit further spread of COVID-19 following its recent emergence, and symptom screening has become a ubiquitous tool in the global response. Previously, we developed a mathematical model to understand factors governing the effectiveness of traveller screening to prevent spread of emerging pathogens (Gostic et al., 2015). Here, we estimate the impact of different screening programs given current knowledge of key COVID-19 life history and epidemiological parameters. Even under best-case assumptions, we estimate that screening will miss more than half of infected people. Breaking down the factors leading to screening successes and failures, we find that most cases missed by screening are fundamentally undetectable, because they have not yet developed symptoms and are unaware they were exposed. Our work underscores the need for measures to limit transmission by individuals who become ill after being missed by a screening program. These findings can support evidence-based policy to combat the spread of COVID-19, and prospective planning to mitigate future emerging pathogens.

**\*For correspondence:**
kgostic@uchicago.edu (KG);
jlloydsmith@ucla.edu (JOL-S)

**Competing interests:** The authors declare that no competing interests exist.

## Introduction

As of February 20, 2020, the 2019 novel coronavirus (now named SARS-CoV-2, causing the disease COVID-19) has caused over 75,000 confirmed cases inside of China and has spread to 25 other countries (*World Health Organization, 2020b*). (HCoV-19 has been proposed as an alternate name for the virus; *Jiang et al., 2020*). Until now, local transmission remained limited outside of China, but as of this week, new epidemic hotspots have become apparent on multiple continents (*World Health Organization, 2020a*; *Jankowicz, 2020*; *Sang-Hun, 2020*; *Schnirring, 2020a*). Many jurisdictions have imposed traveller screening in an effort to prevent importation of COVID-19 cases to unaffected areas. Some high-income countries have escalated control measures beyond screening-based containment policies, and now restrict or quarantine inbound travellers from countries known to be experiencing substantial community transmission. Meanwhile, in many other countries, screening remains the primary barrier to case importation (*Guardian reporting team, 2020*; *Schengen Visa Info, 2020*). Even in countries with the resources to enforce quarantine measures, expanded arrival screening may be the first logical response as the source epidemic expands to regions outside China. Furthermore, symptom screening has become a ubiquitous tool in the effort to contain local spread of COVID-19, in settings from affected cities to cruise ships to quarantines. Our analysis is pertinent to all of these contexts.

It is widely recognized that screening is an imperfect barrier to spread (*Bitar et al., 2009*; *Cowling et al., 2010*; *Gostic et al., 2015*; *Mabey et al., 2014*; *Quilty et al., 2020*), due to: the absence of detectable symptoms during the incubation period; variation in the severity and detectability of symptoms once the disease begins to progress; imperfect performance of screening equipment or personnel; or active evasion of screening by travellers. Previously we estimated the effectiveness of traveller screening for a range of pathogens that have emerged in the past, and found that arrival screening would miss 50–75% of infected cases even under optimistic assumptions (*Gostic et al., 2015*). Yet the quantitative performance of different policies matters for planning interventions and will influence how public health authorities prioritize different measures as the international and domestic context changes. Here we use a mathematical model to analyse the expected performance of different screening measures for COVID-19, based on what is currently known about its natural history and epidemiology and on different possible combinations of departure and arrival screening policies.

First we assess the probability that any single individual infected with SARS-CoV-2 would be detected by screening, as a function of time since exposure. This individual-level analysis is not a comprehensive measure of program success, but serves to illustrate the various ways in which screening can succeed or fail (and in turn the ways it can or cannot be improved). Further, these analyses emphasize the importance of the incubation period, and the fraction of subclinical cases, as determinants of individual screening outcomes. We define subclinical cases as those too mild to show symptoms detectable in screening (fever or cough) after passing through the incubation period (i.e. once any symptoms have manifested). The true fraction of subclinical COVID-19 cases remains unknown, but anecdotally, many lab-confirmed COVID-19 cases have not shown detectable symptoms on diagnosis (*Hoehl et al., 2020*; *Nishiura et al., 2020*; *Hu et al., 2020*). About 80% of clinically attended cases have been mild (*The Novel Coronavirus Pneumonia Emergency Response Epidemiology Team, 2020*), and clinically attended cases have been conspicuously rare in children and teens (*Chen et al., 2020*; *The Novel Coronavirus Pneumonia Emergency Response Epidemiology Team, 2020*; *Huang et al., 2020*; *Li et al., 2020*), raising the possibility that subclinical cases may be common.

Next, we assess the overall effectiveness of a screening program by modeling screening outcomes in a hypothetical population of infected travellers, each with a different time since exposure (and hence a different probability of having progressed through incubation to show detectable symptoms). Crucially, the distribution of times since exposure will depend on the epidemiology of the source population, so this overall measure is not a simple average of the individual-level outcomes. We estimate the fraction of infected travellers detected, breaking down the ways in which screening can succeed or fail. An alternate measure of program success is the extent to which screening delays the first importation of cases into the community, possibly providing additional time to train medical staff, deploy public health responders or refine travel policies (*Cowling et al., 2010*). To quantify the potential for screening to delay case importation, we estimate the probability that a given screening program would detect the first *n* or more imported cases before missing an infected person.

Screening will be less effective in a growing epidemic, due to an excess of recently-exposed and not-yet-symptomatic travellers (*Gostic et al., 2015*). In the context of COVID-19, we consider both growing and stable epidemic scenarios, but place greater emphasis on the realistic assumption that the COVID-19 epidemic is still growing. Since late January 2020, the Chinese government has imposed strict travel restrictions and surveillance on population centers heavily affected by COVID-19 (*BBC News, 2020*; *Cellan-Jones, 2020*), and numerous other countries have imposed travel and quarantine restrictions on travellers inbound from China. Until about Feb. 20, 2020, these measures had appeared to successfully limit community transmission outside of China, but all the while multiple factors pointed to on-going risk, including evidence that transmission is possible prior to the onset of symptoms (*Yu et al., 2020*; *Hu et al., 2020*), and reports of citizens seeking to elude travel restrictions or leaving before restrictions were in place (*Ma and Pinghui, 2020*; *Mahbubani, 2020*). Now, in the week following Feb. 20, 2020, new source epidemics have appeared on multiple

continents (*World Health Organization, 2020a*), and the the risk of exportation of cases from beyond the initial travel-restricted area is growing.

As the epidemic continues to expand geographically, arrival screening will likely be continued or expanded to prevent importation of cases to areas without established spread. At the same time, there is great concern about potential public health consequences if COVID-19 spreads to developing countries that lack health infrastructure and resources to combat it effectively (*de Salazar et al., 2020*). Limited resources also could mean that some countries cannot implement large-scale arrival screening. In this scenario, departure screening implemented elsewhere would be the sole barrier – however leaky – to new waves of case importation. It is also important to recognize that, owing to the lag time in appearance of symptoms in imported cases, any weaknesses in screening would continue to have an effect on known case importations for up to two weeks, officially considered the maximum incubation period (*World Health Organization, 2020c*). Accordingly, we consider scenarios with departure screening only, arrival screening only, or both departure and arrival screening. The model can also consider the consequences when only a fraction of the traveller population is screened, due either to travel from a location not subject to screening, or due to deliberate evasion of screening.

Our analysis also has direct bearing on other contexts where symptom screening is being used, beyond international air travel. This includes screening of travelers at rail stations and roadside spot checks, and screening of other at-risk people including people living in affected areas, health-care workers, cruise ship passengers, evacuees and people undergoing quarantine (*Hoehl et al., 2020*; *Japan Ministry of Health, Labor and Welfare, 2020*; *Nishiura et al., 2020*; *Schnirring, 2020c*). Below, we chiefly frame our findings in terms of travel screening, but these other screening contexts are also in the scope of our analysis. Any one-off screening effort is equivalent to a departure screen (i.e. a single test with no delay), and our findings on symptom screening effectiveness over the course of infection are directly applicable to longitudinal screening in quarantine or occupational settings.

The central aim of our analysis is to assess the expected effectiveness of screening for COVID-19, taking account of current knowledge and uncertainties about the natural history and epidemiology of the virus. We therefore show results using the best estimates currently available, in the hope of informing policy decisions in this fast-changing environment. We also make our model available for public use as a user-friendly online app, so that stakeholders can explore scenarios of particular interest, and results can be updated rapidly as our knowledge of this new viral threat continues to expand.

## Results

### Model for COVID-19 screening

The core model has been described previously (*Gostic et al., 2015*), but to summarize briefly, it assumes infected travellers can be detained due to the presence of detectable symptoms (fever or cough), or due to self-reporting of exposure risk via questionnaires or interviews. These assumptions are consistent with WHO traveller screening guidelines (*World Health Organization, 2020b*; *World Health Organization, 2020c*). Upon screening, travellers fall into one of four categories: (1) symptomatic but not aware of exposure risk, (2) aware of exposure risk but without detectable symptoms, (3) symptomatic and aware that exposure may have occurred, and (4) neither symptomatic nor aware of exposure risk (*Figure 1*). Travellers in the final category are fundamentally undetectable, and travellers in the second category are only detectable if aware that they have been exposed and willing to self report.

In the model, screening for symptoms occurs prior to questionnaire-based screening for exposure risk, and detected cases do not progress to the next stage. This allows us to track the fraction of cases detected using symptom screening or risk screening at arrival or departure. Additionally, building on the four detectability classes explained above, the model keeps track of four ways in which screening can miss infected travellers: (1) due to imperfect sensitivity, symptom screening may fail to detect symptoms in travellers that display symptoms; (2) questionnaires may fail to detect exposure risk in travellers aware they have been exposed, owing to deliberate obfuscation or misunderstanding; (3) screening may fail to detect both symptoms and known exposure risk in travellers

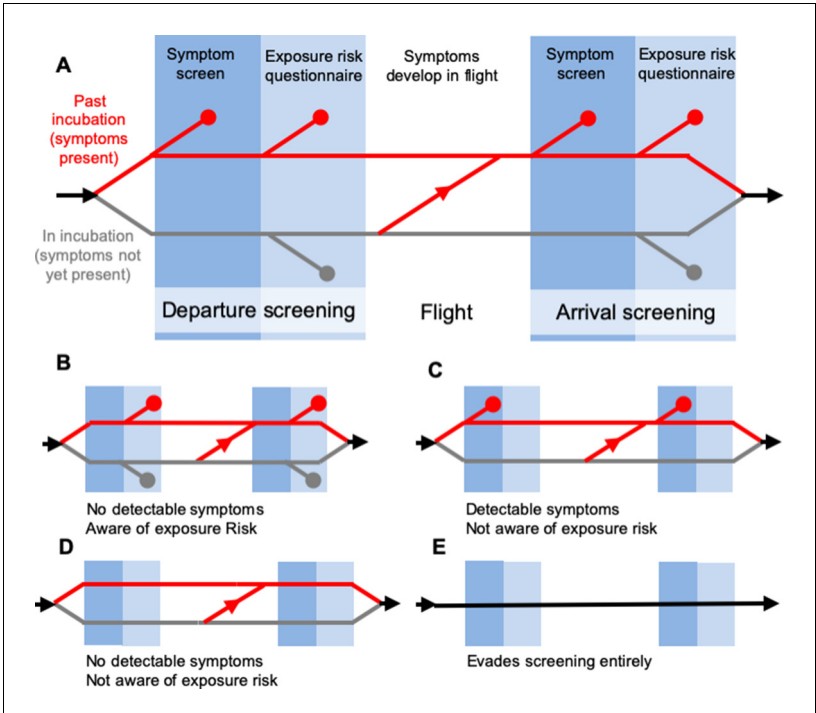

**Figure 1.** Model of traveller screening process, adapted from *Gostic et al. (2015)*. Infected travellers fall into one of five categories: (**A**) Cases aware of exposure risk and with fever or cough are detectable in both symptom screening and questionnaire-based risk screening. (**B**) Cases aware of exposure risk, but without fever or cough are only detectable using risk screening. (**C**) Cases with fever or cough, but unaware of exposure to SARS-CoV-2 are only detectable in symptom screening. (**D–E**) Subclinical cases who are unaware of exposure risk, and individuals that evade screening, are fundamentally undetectable.

who have both and (4) travellers not exhibiting symptoms and with no knowledge of their exposure are fundamentally undetectable. Here, we only consider infected travellers who submit to screening. However, the supplementary app allows users to consider scenarios in which some fraction of infected travellers intentionally evade screening (*Figure 1E*).

The probability that an infected person is detectable in a screening program depends on: the incubation period (the time from exposure to onset of detectable symptoms); the proportion of sub-clinical cases (mild cases that lack fever or cough); the sensitivity of thermal scanners used to detect fever; the fraction of cases aware they have high exposure risk; and the fraction of those cases who would self-report truthfully on a screening questionnaire. Further, the distribution of individual times since exposure affects the probability that any single infected traveller has progressed to the symptomatic stage. If the source epidemic is still growing, the majority of infected cases will have been recently exposed, and will not yet show symptoms. If the source epidemic is no longer growing (stable), times since exposure will be more evenly distributed, meaning that more infected travellers will have progressed through incubation and will show detectable symptoms.

We used methods described previously to estimate the distribution of individual times since exposure in a growing or stable epidemic, given various values of the reproductive number $R_0$ (*Gostic et al., 2015*). Briefly, early in the epidemic when the number of cases is still growing, the model draws on epidemiological theory to assume that the fraction of cases who are recently exposed increases with $R_0$. The distribution of times since exposure is truncated at a maximum value, which corresponds epidemiologically to the maximum time from exposure to patient isolation, after which point we assume cases will not attempt to travel. (Isolation may occur due to hospitalization, or due to confinement at home in response to escalating symptoms or COVID-19 diagnosis. In the non-travel context, this would correspond to cases that have been hospitalized or otherwise diagnosed and isolated.) Here, we approximate the maximum time from exposure to isolation as the sum of the mean incubation time, and mean time from onset to isolation. To consider the

epidemiological context of a stable epidemic in the source population we assume times since exposure follow a uniform distribution across the time period between exposure and isolation.

## Parameters, uncertainty and sensitivity analyses

As of February 20, 2020, COVID-19-specific estimates are available for most parameters, but many have been derived from limited or preliminary data and remain subject to considerable uncertainty. *Table 1* and the Methods summarize the current state of knowledge. Here, we used two distinct approaches to incorporate parameter uncertainty into our analysis.

First, to estimate the probability that an infected individual would be detected or missed we considered a range of plausible values for the mean incubation time, and the fraction of subclinical

**Table 1.** Parameter values estimated in currently available studies, along with accompanying uncertainties and assumptions. Ranges in the final column reflect confidence interval, credible interval, standard error or range reported by each study referenced.

| Parameter | Best estimate (Used in *Figure 2*) | Plausible range (Used in *Figure 3*) | References and notes |
|---|---|---|---|
| Mean incubation period | 5.5 days Sensitivity: 4.5 or 6.5 days | 4.5–6.5 days | 3–6 days, n = 4 (*Chan et al., 2020*)* 5.2 (4.1–7.0) days, n < 425 (*Li et al., 2020*)[†] 5.2 (4.4–6.0) days, n = 101 (*Lauer et al., 2020*)[†] 6.5 (5.6–7.9) days, n = 88 (*Backer et al., 2020*)[†] |
| Incubation period distribution | Gamma distribution with mean as above, and standard deviation = 2.25 | | |
| Percent of cases subclinical (No fever or cough) | Best case scenario: 5% Middle case scenario: 25% Worst case scenario: 50% | | Clinical data: 83% fever, 67% cough, n = 6 (*Chan et al., 2020*) 83% fever, 82% cough, n = 99 (*Chen et al., 2020*) 98% fever, 76% cough, n = 41 (*Huang et al., 2020*) 43.8% fever at hospital admission, 88.7% fever during hospitalization, n = 1099 (*Guan et al., 2020*) Active monitoring after repatriation flights or on cruise ships: % asymptomatic at diagnosis 31.2% (111/355) (*Japan Ministry of Health, Labor and Welfare, 2020*) 65.2% (5 of 8) (*Nishiura et al., 2020*) 70.0% (7 of 10) (*Dorigatti et al., 2020*) |
| $R_0$ | No effect in individual-level analysis. | 1.5–4.0 | 2.2 (1.4–3.8) (*Riou and Althaus, 2020*) 2.2 (1.4–3.9) (*Li et al., 2020*) 2.6 (1.5–3.5) (*Imai et al., 2020*) 2.7 (2.5–2.9) (*Wu et al., 2020*) 4.5 (4.4-4.6) (*Liu et al., 2020*) 3.8 (3.6-4.0) (*Read et al., 2020*) 4.08 (3.37–4.77) (*Cao et al., 2020*) 4.7 (2.8–7.6) (*Sanche et al., 2020*) 6.3 (3.3-11.3) (*Sanche et al., 2020*) 6.47 (5.71–7.23) (*Tang et al., 2020*) |
| Percent of travellers aware of exposure risk | 20% | 5–40% | We assume a low percentage, as no specific risk factors have been identified, and known times or sources of exposure are rarely reported in existing line lists. |
| Sensitivity of infrared thermal scanners for fever | 70% | 60–90% | Most studies estimated sensitivity between 60–88% (*Bitar et al., 2009*; *Priest et al., 2011*; *Tay et al., 2015*). But a handful of studies estimated very low sensitivity (4–30%). In general, sensitivity depended on the device used, body area targeted and ambient temperature. |
| Probability that travellers self-report exposure risk | 25% | 5–25% | 25% is an upper-bound estimate based on outcomes of past screening initiatives. (*Gostic et al., 2015*) |
| Time from symptom onset to patient isolation (After which we assume travel is not possible) | No effect in individual-level analysis. | 3–7 days | Median 7 days from onset to hospitalization (n = 6) (*Chan et al., 2020*) Mean 2.9 days onset to patient isolation (n = 164) (*Liu et al., 2020*) Median 7 days from onset to hospitalization (n = 41) (*Huang et al., 2020*) As awareness increases, times to isolation may decline. |

\* From family cluster.

† Parametric distributions fit to cases with known dates of exposure or travel to and from Wuhan.

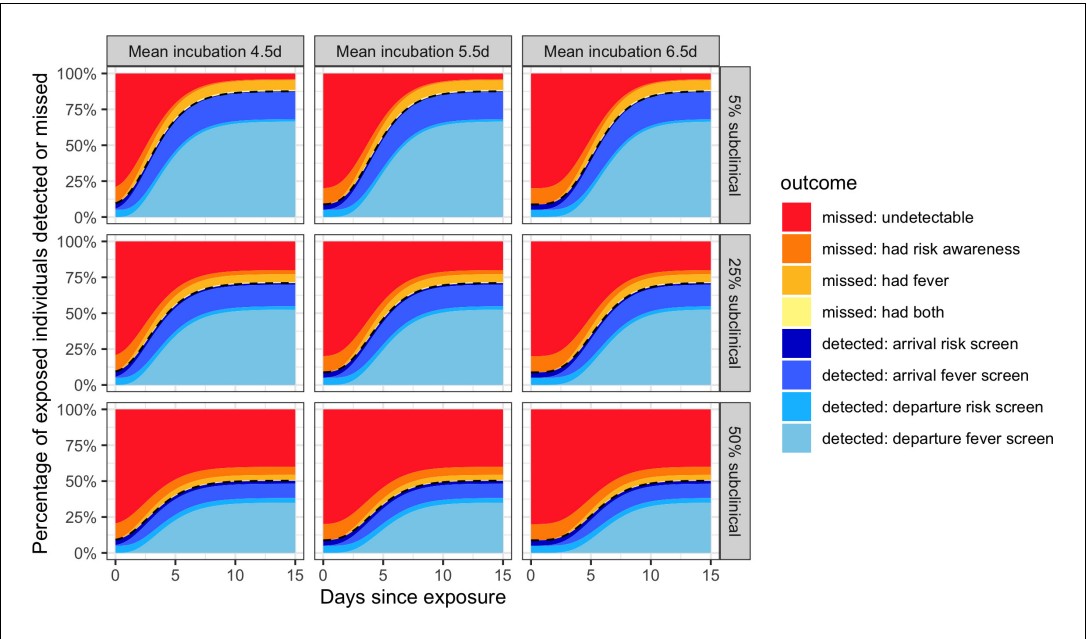

**Figure 2.** Individual outcome probabilities for travellers who screened at given time since infection. Columns show three possible mean incubation periods, and rows show best-case, middle-case and worst-case estimates of the fraction of subclinical cases. Here, we assume screening occurs at both arrival and departure; see *Figure 2—figure supplement 1* and *Figure 2—figure supplement 2* for departure or arrival screening only. The black dashed lines separate detected cases (below) from missed cases (above). Here, we assume flight duration = 24 hr, the probability that an individual is aware of exposure risk is 0.2, the sensitivity of fever scanners is 0.7, and the probability that an individual will truthfully self-report on risk questionnaires is 0.25. *Table 1* lists all other input values.

The online version of this article includes the following source data and figure supplement(s) for figure 2:

**Source data 1.** Source data for *Figure 2*.
**Figure supplement 1.** Departure screening only.
**Figure supplement 2.** Arrival screening only.

cases. We focus on the incubation period and subclinical fraction of cases because screening outcomes are particularly sensitive to their values. All other parameters were fixed to the best available estimates listed in *Table 1*.

Second, we considered a population of infected travellers, each with a unique time of exposure, and in turn a unique probability of having progressed to the symptomatic stage. Here, the model used a resampling-based approach to simultaneously consider uncertainty from both stochasticity in any single individual's screening outcome, and uncertainty as to the true underlying natural history parameters driving the epidemic. Details are provided in the methods, but briefly, we constructed 1000 candidate parameter sets, drawn using Latin hypercube sampling from plausible ranges for each parameter (*Table 1*). Using each parameter set, we simulated one set of screening outcomes for a population of 30 infected individuals. As of February 20, 2020, 30 approximates the maximum known number of cases imported to any single country (*World Health Organization, 2020b*), and thus our analysis incorporates a reasonable degree of binomial uncertainty. The actual number of infected travellers passing through screening in any given location may be higher or lower than 30, and will depend on patterns of global connectivity, and on the duration of the source epidemic (*Chinazzi et al., 2020*; *de Salazar et al., 2020*). Finally, we analysed the sensitivity of screening effectiveness (fraction of travellers detected) to each parameter, as measured by the partial rank correlation coefficient (PRCC) (*Marino et al., 2008*).

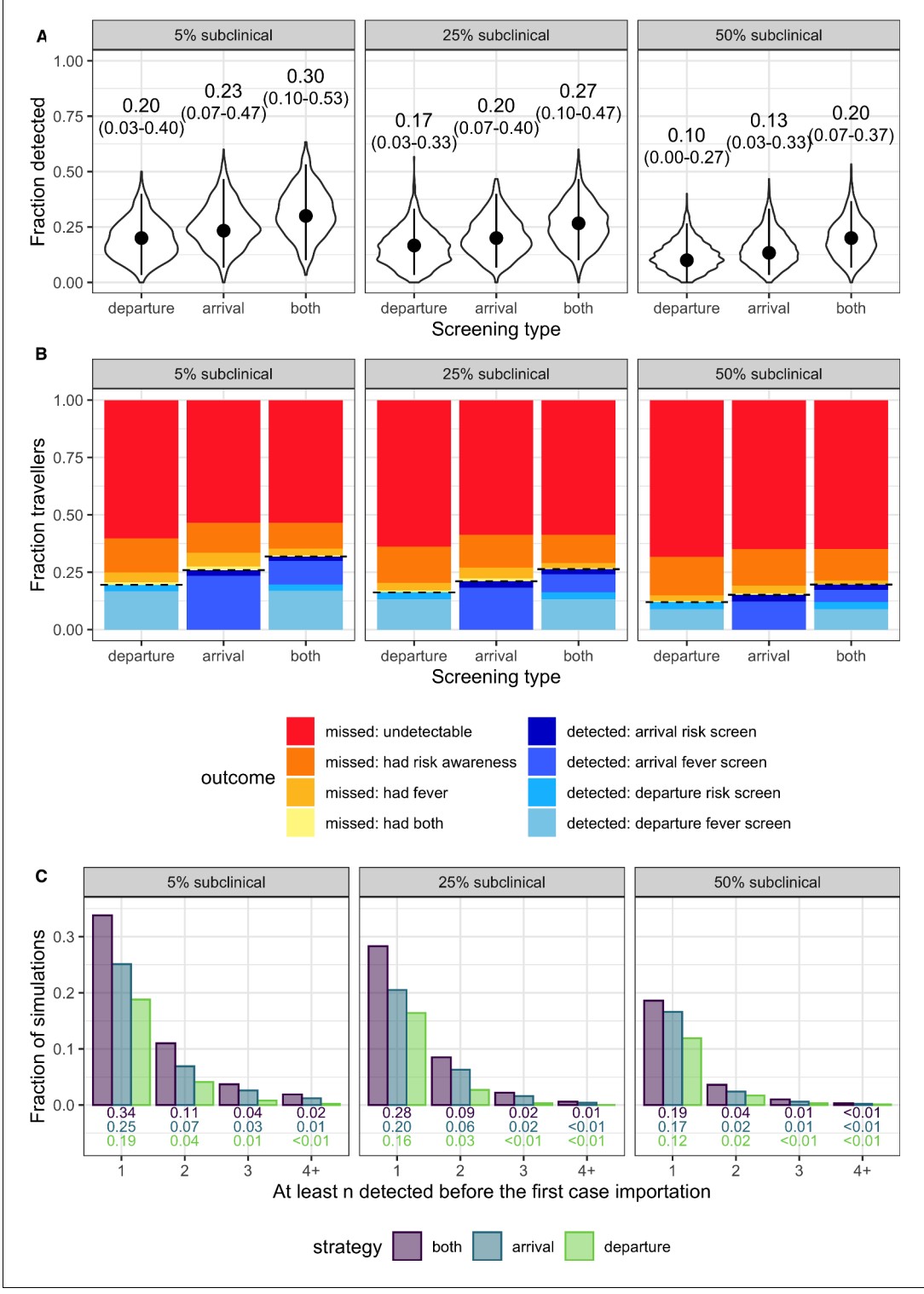

**Figure 3.** Population-level outcomes of screening programs in a growing epidemic. (**A**) Violin plots of the fraction of infected travellers detected, accounting for current uncertainties by running 1000 simulations using parameter sets randomly drawn from the ranges shown in **Table 1**. Dots and vertical line segments show the median and central 95%, respectively. Text above each violin shows the median and central 95% fraction detected. (**B**) Mean fraction of travellers with each screening outcome. The black dashed lines separate detected cases (below) from missed cases (above). (**C**) Fraction of simulations in which screening successfully detects at least n cases before the first infected traveller is missed.

*Figure 3 continued on next page*

*Figure 3 continued*

The online version of this article includes the following source data and figure supplement(s) for figure 3:

**Source data 1.** Source data for *Figure 3A*.
**Source data 2.** Source data for *Figure 3B*.
**Source data 3.** Source data for *Figure 3C*.
**Figure supplement 1.** Population-level screening outcomes given that the source epidemic is no longer growing.
**Figure supplement 2.** Plausible incubation period distributions underlying the analyses in *Figure 3*.

## Individual probabilities of detection

First, the model estimated the probability that any single infected individual would be detected by screening as a function of the time between exposure and the initiation of travel (*Figure 2*). Incubation time is a crucial driver of traveller screening effectiveness; infected people are most likely to travel before the onset of symptoms. Here we considered three mean incubation times, which together span the range of most existing mean estimates: 4.5, 5.5 and 6.5 days (*Table 1*). Additionally, we considered three possible fractions of subclinical cases: 50% represents a worst-case upper bound, 5% represents a best-case lower bound, and 25% represents a plausible middle case. (*Table 1*, Materials and methods). Since resubmission, a new delay-adjusted estimate indicates that 34.6% of infections are asymptomatic (*Mizumoto et al., 2020*), intermediate between our middle and worst-case scenarios.

Even within the narrow range tested, screening outcomes were sensitive to the incubation period mean. For longer incubation periods, we found that larger proportions of departing travellers would not yet be exhibiting symptoms – either at departure or arrival – which in turn reduced the probability that screening would detect these cases, especially since we assume few infected travellers will realize they have been exposed to COVID-19.

A second crucial uncertainty is the proportion of subclinical cases, which lack detectable fever or cough even after the onset of symptoms. We considered scenarios in which 5%, 25% and 50% of cases are subclinical, representing a best, middle and worst-case scenario, respectively. The middle and worst-case scenarios have predictable and discouraging consequences for the effectiveness of traveller screening, since they render large fractions of the population undetectable by fever screening (*Figure 2*). Furthermore, subclinical cases who are unaware of their exposure risk are never detectable, by any means. This is manifested as the bright red 'undetectable' region which persists well beyond the mean incubation period. For a screening program combining departure and arrival screening, as shown in *Figure 2*, the greatest contributor to case detection is the departure fever screen. The arrival fever screen is the next greatest contributor, with its value arising from two factors: the potential to detect cases whose symptom onset occurred during travel, and the potential to catch cases missed due to imperfect instrument sensitivity in non-contact infrared thermal scanners used in traveller screening (*Table 1*). Considering the effectiveness of departure or arrival screening only (*Figure 2—figures supplement 1*, *2*), we see that fever screening is the dominant contributor in each case, but that the risk of missing infected travellers due to undetected fever is substantially higher when there is no redundancy from two successive screenings.

## Overall screening effectiveness in a population of infected travellers during a growing or stable epidemic

Next we estimated the overall effectiveness of different screening programs, as well as the uncertainties arising from the current partial state of knowledge about this recently-emerged virus. We modeled plausible population-level outcomes by tracking the fraction of 30 infected travellers detained, given a growing or stable epidemic and current uncertainty around parameter values. We separately consider the best, middle and worst-case scenarios for the proportion of infections that are subclinical, and for each scenario we compare the impact of departure screening only (or equivalently, any on-the-spot screening), arrival screening only, or programs that include both.

The striking finding is that in a growing epidemic, even under the best-case assumptions, with just one infection in twenty being subclinical and all travellers passing through departure and arrival screening, the median fraction of infected travellers detected is only 0.30, with 95% interval extending from 0.10 up to 0.53 (*Figure 3A*). The total fraction detected is lower for programs with only

one layer of screening, with arrival screening preferable to departure screening owing to the possibility of symptom onset during travel. Considering higher proportions of subclinical cases, the overall effectiveness of screening programs is further degraded, with a median of just one in ten infected travellers detected by departure screening in the worst-case scenario. The key driver of these poor outcomes is that even in the best-case scenario, nearly two thirds of infected travellers will not be detectable (as shown by the red regions in *Figure 3B*). There are three drivers of this outcome: (1) in a growing epidemic, the majority of travellers will have been recently infected and hence will not yet have progressed to exhibit any symptoms; (2) we assume that a fraction of cases never develop detectable symptoms; and (3) we assume that few people are aware of their exposure risk. As above, the dominant contributor to successful detections is fever screening.

In an epidemic that is no longer growing (*Figure 3—figures supplement 1*), screening effectiveness is considerably higher, as a lower proportion of travellers will be recently exposed. This is shown by the smaller, red 'undetectable' region in *Figure 3—figures supplement 1B*. In a stable epidemic, under the middle-case assumption that 25% of cases are subclinical, we estimate that arrival screening alone would detect roughly one third (17–53%) of infected travelers, and that a combination of arrival and departure screening would detect nearly half (23–63%) of infected travellers. In short, holding all other things equal, screening effectiveness will increase as the source epidemic transitions from growing to stable, owing simply to changes in the distribution of 'infection ages,' or times since exposure.

To assess the potential for screening to delay introduction of undiagnosed cases, we evaluated the fraction of simulations in which screening during a growing epidemic would detect the first *n* or more infected travellers (*Figure 3C*). Depending on the screening strategy (arrival, departure or both) and assumed subclinical fraction (5%, 25%, or 50%), the probability of detecting at least the first two cases ranged from 0.02 to 0.11, and the probability of detecting three or more cases was never better than 0.04 (*Figure 3C*). In all tested scenarios, more than half of simulations failed to detect the first imported case, consistent with probabilities of case detection in *Figure 3A*. Probabilities of detecting the first *n* consecutive cases were marginally higher in the stable epidemic context (*Figure 3—figures supplement 1*), but still the probability of detecting at least the first three cases was never better than 0.13, and the probability of detecting the first four cases was never better than 0.06 in any tested scenario. Taken together, these results indicate that screening in any context is very unlikely to delay case importation beyond the first 1–3 cases, and often will not delay the first importation at all. What duration of delay this yields will depend on the frequency of infected travellers.

## Sensitivity analysis

In the context of a growing epidemic, sensitivity analysis using the method of Latin hypercube sampling and partial rank correlation (*Marino et al., 2008*) showed that the fraction of travellers detected was moderately sensitive to all parameters considered – most coefficient estimates fell between 0.1 and 0.3 in absolute value (*Figure 4*). Sensitivity to $R_0$ was somewhat higher than sensitivity to other parameters, but the difference was not statistically remarkable. $R_0$ and the mean incubation period were negatively associated with the fraction of cases detected. An increase in either of these parameters implies an increase in the probability an infected traveller will be undetectable, either because they have been recently exposed ($R_0$), or have not yet progressed to the symptomatic stage (mean incubation time). The positive association between the fraction of cases detected and the sensitivity of thermal scanners, sensitivity of risk questionnaires, or the fraction of travellers aware of exposure risk is intuitive. Finally, the duration from onset to isolation effectively describes the window of time in which we assume a symptomatic individual could initiate travel. Here, a wider window is associated with increased screening effectiveness, because it will lead to a higher proportion of infected travellers who are symptomatic. *Figure 4* shows results from the middle case scenario, in which 25% of cases are subclinical. Considering scenarios where more or fewer cases are subclinical, we see increased influence of the factors based on exposure risk (questionnaire sensitivity and the fraction of cases aware of their exposure) as the proportion of cases with detectable symptoms declines (*Figure 4—figures supplement 1*).

In the context of a stable epidemic, a greater proportion of infected travellers will have progressed to show detectable symptoms, and so screening effectiveness was more sensitive to parameters that impact symptom screening efficacy (thermal scanner sensitivity, and to the time from

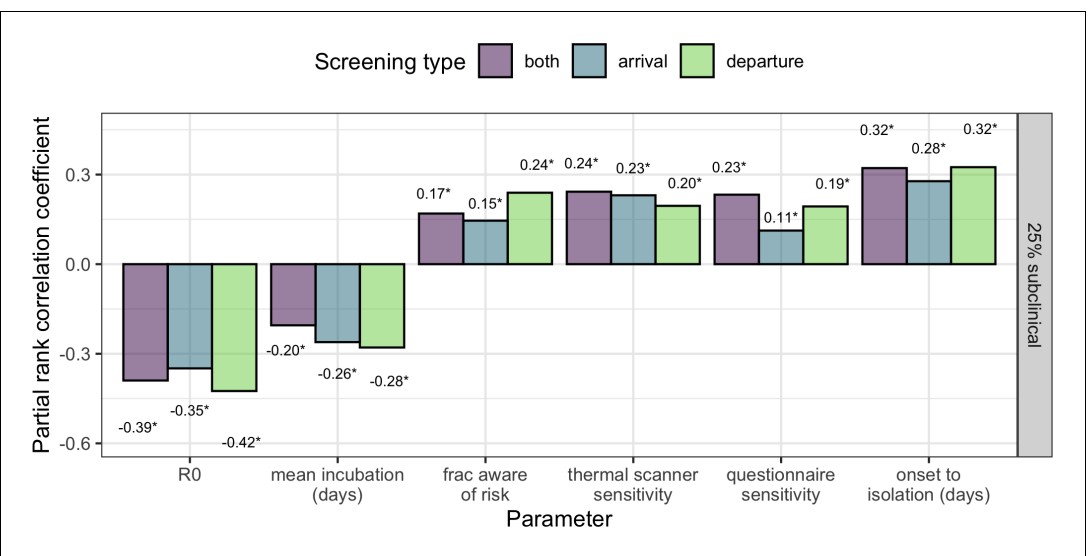

**Figure 4.** Sensitivity analysis showing partial rank correlation coefficient (PRCC) between each parameter and the fraction (per-simulation) of 30 infected travellers detected. Outcomes were obtained from 1000 simulations, each using a candidate parameter sets drawn using Latin hypercube sampling. Text shows PRCC estimate, and * indicates statistical significance after Bonferroni correction (threshold = 9e-4 for 54 comparisons).

The online version of this article includes the following source data and figure supplement(s) for figure 4:

**Source data 1.** Source data for *Figure 4*, and *Figure 4—figures supplement 1*.
**Figure supplement 1.** PRCC analysis comparing cases where 5%, 25% or 50% of cases are subclinical.
**Figure supplement 2.** PRCC analysis assuming the source epidemic is no longer growing.

symptom onset to isolation). Note that by construction, model outcomes are insensitive to parameter $R_0$ in the stable epidemic context. As a result, $R_0$ coefficient estimates are very small (non-zero due to stochasticity in simulation outcomes), and never significant. (*Figure 4—figures supplement 2*).

## Interactive online app for public use

We have developed an interactive web application using the R package Shiny (*Chang et al., 2019*) in which users can replicate our analyses using parameter inputs that reflect the most up-do-date information. The supplementary user interface can be accessed at https://faculty.eeb.ucla.edu/lloyd-smith/screeningmodel. Please note that while the results in *Figures 3* and *4* consider a range of plausible values for each parameter, the outputs of the Shiny app are calculated using fixed, user-specified values only.

## Discussion

The international expansion of COVID-19 cases has led to widespread adoption of symptom and risk screening measures, in travel-associated and other contexts, and programs may still be adopted or expanded as source epidemics of COVID-19 emerge in new geographic areas. Using a mathematical model of screening effectiveness, with preliminary estimates of COVID-19 epidemiology and natural history, we estimate that screening will detect less than half of infected travellers in a growing epidemic, and that screening effectiveness will increase marginally as growth of the source epidemic decelerates. We found that two main factors influenced the effectiveness of screening. First, symptom screening depends on the natural history of an infection: individuals are increasingly likely to show detectable symptoms with increasing time since exposure. A fundamental shortcoming of screening is the difficulty of detecting infected individuals during their incubation period, or early after the onset of symptoms, at which point they still feel healthy enough to undertake normal activities or travel. This difficulty is amplified when the incubation period is longer; infected individuals have a longer window in which they may mix socially or travel with low probability of detection.

Second, screening depends on whether exposure risk factors exist that would facilitate specific and reasonably sensitive case detection by questionnaire. For COVID-19, there is so far limited evidence for specific risk factors; we therefore assumed that at most 40% of travellers would be aware of a potential exposure. It is plausible that many individuals aware of a potential exposure would voluntarily avoid travel and practice social distancing–if so, the population of infected travellers will be skewed toward those unaware they have been exposed. Furthermore, based on screening outcomes during the 2009 influenza pandemic, we assumed that a minority of infected travellers would self-report their exposure honestly, which led to limited effectiveness in questionnaire-based screening. The confluence of these two factors led to many infected people being fundamentally undetectable in our model. Even under our most generous assumptions about the natural history of COVID-19, the presence of undetectable cases made the greatest contribution to screening failure. Correctable failures, such as missing an infected person with fever or awareness of their exposure risk, played a more minor role.

Our conclusion that screening would detect no more than half of infected travellers in a growing epidemic is consistent with recent studies that have compared country-specific air travel volumes with detected case counts to estimate that roughly two thirds of imported cases remain undetected (*Niehus et al., 2020*; *Bhatia et al., 2020*). Furthermore, the finding that the majority of cases missed by screening are fundamentally undetectable is consistent with observed outcomes so far. Analyzing a line list of 290 cases imported into various countries (*Dorigatti et al., 2020*), we found that symptom onset occurred after the date of inbound travel for 72% (75/104) of cases for whom both dates were available, and a further 14% (15/104) had symptom onset on the date of travel. Even among passengers of repatriation flights, or quarantined on a cruise ship off the coast of Japan (who are all demonstrably at high risk), numerous cases have been undetectable in symptom screening, but have still tested positive for SARS-CoV-2 by PCR (*Dorigatti et al., 2020*; *Hoehl et al., 2020*; *Japan Ministry of Health, Labor and Welfare, 2020*; *Nishiura et al., 2020*; *Hu et al., 2020*). The onset of viral shedding prior to the onset of symptoms, or in cases that remain asymptomatic, is a classic factor that makes infectious disease outbreaks difficult to control (*Fraser et al., 2004*).

Our results emphasize that the true fraction of subclinical cases (those who lack fever or cough at symptom onset) remains a crucial unknown, and strongly impacts screening effectiveness. Reviewing data from active surveillance of passengers on cruise ships or repatriation flights, we estimate that up to half of cases show no detectable symptoms at the time of diagnosis. To complicate matters further, the fraction of subclinical cases may vary across age groups. Children and young adults have been conspicuously underrepresented, even in very large clinical data sets (*Chen et al., 2020*; *The Novel Coronavirus Pneumonia Emergency Response Epidemiology Team, 2020*; *Huang et al., 2020*; *Li et al., 2020*). Only 2.1% of the first 44,672 confirmed cases were observed in children under 20 years of age (*The Novel Coronavirus Pneumonia Emergency Response Epidemiology Team, 2020*). The possibility cannot be ruled out that large numbers of subclinical cases are occurring in young people. If an age-by-severity interaction does indeed exist, then the mean age of travellers should be taken into account when estimating screening effectiveness.

There are some limitations to our analysis. Parameter values for COVID-19 may be affected by bias or censoring, particularly in the early stages of an outbreak when most cases have been recently infected, and when severe or hospitalized cases are overrepresented in the available data. In particular, the tail of the incubation period distribution is difficult to characterize with precision using limited or biased data.

As country-specific screening policies can change rapidly in real-time, we focused on a general screening framework rather than specific case studies. We also assumed traveller adherence and no active evasion of screening. However, there are informal reports of people taking antipyretics to beat fever screening (*Mahbubani, 2020*), which would further reduce the effectiveness of these methods. With travel restrictions in place, individuals may also take alternative routes (e.g. road rather than air), which would in effect circumvent departure and/or arrival screening as a control measure. Our quantitative findings may overestimate screening effectiveness if many travellers evade screening.

Our results have several implications for the design and implementation of traveller screening policies. If the infection is not yet present in a region, then arrival screening could delay the introduction of cases, but consistent with previous analyses, (*Cowling et al., 2010*), our results indicate such delays would be minimal. Our findings indicate that for every case detected by travel screening, one

or more infected travellers were not caught, and must be found and isolated by other means. We note that even with high $R_0$ and no control measures in place, a single case importation is not guaranteed to start a sustained chain of transmission (*Kucharski et al., 2020*; *Lloyd-Smith et al., 2005*). This is particularly true for infections that exhibit a tendency toward superspreading events, as increasingly reported for COVID-19, but the flipside is that outbreaks triggered by superspreading are explosive when they do occur (*Lloyd-Smith et al., 2005*).

We did not analyze second-order benefits from screening, such as potential to raise awareness. Official recommendations emphasize that screening is an opportunity for 'risk communication' in which travellers can be instructed how to proceed responsibly if symptoms develop at the destination (*World Health Organization, 2020d*). Alongside increased general surveillance/alertness in healthcare settings, such measures could help reduce the risk of local transmission and superspreading, but their quantitative effectiveness is unknown. Once limited local transmission has begun, arrival screening could still have merit as a means to restrict the number of parallel chains of transmission present in a country. Once there is generalized spread which has outpaced contact tracing, departure screening to prevent export of cases to new areas will be more valuable than arrival screening to identify additional incoming cases. Altogether, screening should not be viewed as a definitive barrier to case importation, but used alongside on-the-ground response strategies that help reduce the probability that any single imported case spreads to cause a self-sustaining local epidemic. The cost-benefit tradeoff for any screening policy should be assessed in light of past experiences, where few or no infected travellers have been detected by such programs (*Gostic et al., 2015*).

While our findings indicate that the majority of screening failures arise from undetectable cases (i.e. those without symptoms or knowledge of their exposure), several factors could potentially strengthen the screening measures described here. With improved efficiency of thermal scanners or other symptom detection technology, we would expect a smaller difference between the effectiveness of arrival-only screening and combined departure and arrival screening in our analysis. Alternatively, the benefits of redundant screening (noted above for programs with departure and arrival screens) could be gained in a single-site screening program by simply having two successive fever-screening stations that travellers pass through (or taking multiple measurements of each traveller at a single station). As risk factors become better known, questionnaires could be refined to identify more potential cases. Alternatively, less stringent definition of high exposure risk (e.g. contact with anyone with respiratory symptoms) would be more sensitive, but at the expense of large numbers of false positives detained, especially during influenza season.

The availability of rapid PCR tests would also be beneficial for case identification at arrival, and would help address concerns with false-positive detections by screening. If such tests were fast, there may be potential to test suspected cases in real time based on questionnaire responses, travel origin, or borderline symptoms; at least one PCR test for SARS-CoV-2 claimed to take less than an hour has already been announced (*Biomeme, 2020*). However, such measures could prove highly expensive if implemented at scale. There is also scope for new tools to improve the ongoing tracking of travellers who pass through screening, such as smartphone-based self-reporting of temperature or symptoms in incoming cases (*Dorigatti et al., 2020*). Smartphone or diary-based surveillance would be cheaper and more scalable than intense, on-the-ground follow-up, but is likely to be limited by user adherence.

Our analysis underscores the reality that respiratory viruses are difficult to detect by symptom and risk screening programs, particularly if a substantial fraction of infected people show mild or indistinct symptoms, if incubation periods are long, and if transmission is possible before the onset of symptoms. Quantitative estimates of screening effectiveness for COVID-19 will improve as more is learned about this recently-emerged virus, and will vary with the precise design of screening programs. However, we present a robust qualitative finding: in any situation where there is widespread epidemic transmission in source populations from which travellers are drawn, travel screening programs can slow (marginally) but not stop the importation of infected cases. Screening programs implemented in other settings will face the same challenges. By decomposing the factors leading to success or failure of screening efforts, our work supports decision-making about program design, and highlights key questions for further research. We hope that these insights may help to mitigate the global impacts of COVID-19 by guiding effective decision-making in both high- and low-resource

countries, and may contribute to prospective improvements in screening policy for future emerging infections.

## Materials and methods

### Modeling strategy

The model's structure is summarized above (*Figure 1*), and detailed methods have been described previously (*Gostic et al., 2015*). Here, we summarize relevant extensions, assumptions and parameter inputs.

### Extensions

Our previous model tracked all the ways in which infected travellers can be detected by screening (fever screen, or risk factor screen at arrival or departure). Here, we additionally keep track of the many ways in which infected travellers can be missed (i.e. missed given fever present, missed given exposure risk present, missed given both present, or missed given undetectable). Cases who have not yet passed the incubation period are considered undetectable by fever screening, even if they will eventually develop symptoms in the future. In other words, no traveller is considered 'missed given fever present' until they have passed the incubation period and show detectable symptoms. Infected travellers who progress to symptoms during their journey are considered undetectable by departure screening, but detectable by arrival screening.

Additionally, we now provide a supplementary user interface, which allows stakeholders to test input parameters of interest using up-to-date information. Here, in addition to the analyses presented in this study, we implemented the possibility that some fraction of infected travellers deliberately evade screening.

### Basic reproduction number, $R_0$

Existing point estimates for $R_0$ span a wide range (2.2–6.47), but most fall between 2.0 and 4.0 (*Table 1*). The vast majority of these estimates are informed by data collected very early in the outbreak, before any control measures were in place. However, several studies already demostrate decreases in the reproductive number over time, as a consequence of social distancing behaviors, and containment measures (*Kucharski et al., 2020*; *Liu et al., 2020*). Realistically, $R_0$ will vary considerably over time, and across locations, depending on the social context, resource availability, and containment policies. Our analysis considers a plausible range of $R_0$ values spanning 1.5–4.0, with 4.0 representing a plausible maximum in the absence of any behavioral changes or containment efforts, and 1.5 reflecting a plausible lower bound, given containment measures may already be in place at the time of introduction.

### Fraction of subclinical cases

To estimate the upper-bound fraction of subclinical cases, we draw on data from active surveillance of passengers quarantined on a cruise ship off the coast of Japan, or passengers of repatriation flights. These data show that 50–70% of cases are asymptomatic at the time of diagnosis (*Dorigatti et al., 2020*; *Nishiura et al., 2020*; *Schnirring, 2020c*; *Schnirring, 2020b*). We estimate that 50% subclinical cases is a reasonable upper bound: due to intensive monitoring, cases in repatriated individuals or in cruise ship passengers will be detected earlier than usual in the course of infection–and possibly before the onset of symptoms. From clinical data (where severe cases are likely overrepresented), we estimate a lower bound of 5%: even among clinically attended cases, 2–15% lack fever or cough, and would be undetectable in symptom screening (*Chan et al., 2020*; *Chen et al., 2020*; *The Novel Coronavirus Pneumonia Emergency Response Epidemiology Team, 2020*; *Huang et al., 2020*). In addition to the upper and lower bound scenarios, we consider a plausible middle-case scenario in which 25% of cases are subclinical. A very recent delay-adjusted estimate indicates 30-40% of infections on the cruise ship quarantined off the coast of Japan are asymptomatic, so the truth may fall somewhere between our middle and worst-case scenarios (*Mizumoto et al., 2020*).

## Incubation period distribution

We use a gamma distribution to model individual incubation times. We choose this form over the Weibull and lognormal distribution for ease of interpretation (gamma shape and scale parameters can be transformed easily to mean and standard deviation). So far, best-fit gamma distributions to COVID-19 data have had mean 6.5 and standard deviation 2.6 (*Backer et al., 2020*), or mean 5.46 and standard deviation 1.94 (*Lauer et al., 2020*). Here, to model uncertainty around the true mean incubation time, we fix the standard deviation to 2.25 (intermediate between the two existing estimates), and allow the mean to vary between 4.5 and 6.5 days (*Figure 2—figure supplement 2*). The 95th percentile of the distributions we consider fall between 8.7 and 10.6 days, slightly below the officially accepted maximum incubation time of 14 days, and consistent with existing estimates (*Table 1*; *Backer et al., 2020*; *Lauer et al., 2020*).

## Effectiveness of exposure risk questionnaires

The probability that an infected traveller is detectable using questionnaire-based screening for exposure risk will be highest if risk factors with high sensitivity and specificity are known. Currently, official guidance recommends asking whether travellers have visited a country with epidemic transmission, a healthcare facility with confirmed cases, or had close contact with a confirmed or suspected case (*World Health Organization, 2020c*). Given the relative anonymity of respiratory transmission, we assume that a minority of infected travellers would realize that they have been exposed before symptoms develop (20% in *Figure 2*, range 5–40% in *Figure 3*). Further, relying on a previous upper-bound estimate (*Gostic et al., 2015*) we assume that only 25% of travellers would self-report truthfully if aware of elevated exposure risk.

*Table 1* summarizes the state of knowledge about additional parameters, as of February 20, 2020.

## Code and data availability

All code and source data used to perform analyses and generate figures is archived at https://github.com/kgostic/traveller_screening/releases/tag/v2.1. (*Gostic, 2020*; copy archived at https://github.com/elifesciences-publications/traveller_screening).

# Acknowledgements

We are grateful to Miaka McClenahan and Pieter de Ganon for translation assistance. We thank the Cobey lab for helpful comments.

# Additional information

### Funding

| Funder | Grant reference number | Author |
|---|---|---|
| James S. McDonnell Foundation | Postdoctoral fellowship in dynamic and multiscale systems | Katelyn Gostic |
| Wellcome | 206250/Z/17/Z | Adam J Kucharski |
| Coordenação de Aperfeiçoamento de Pessoal de Nível Superior | Science without borders fellowship | Ana CR Gomez |
| National Science Foundation | DEB-1557022 | James O Lloyd-Smith |
| Defense Advanced Research Projects Agency | PREEMPT D18AC00031 | James O Lloyd-Smith |
| Strategic Environmental Research and Development Program | RC-2635 | Ana C R Gomez Riley O Mummah James O Lloyd-Smith |

The funders had no role in study design, data collection and interpretation, or the decision to submit the work for publication.

## Author contributions
Katelyn Gostic, Conceptualization, Resources, Data curation, Software, Formal analysis, Funding acquisition, Investigation, Visualization, Methodology, Writing - original draft, Project administration, Writing - review and editing; Ana CR Gomez, Resources, Software, Validation, Visualization, Methodology, Project administration, Writing - review and editing; Riley O Mummah, Data curation, Project administration, Writing - review and editing; Adam J Kucharski, Conceptualization, Visualization, Writing - original draft, Project administration; James O Lloyd-Smith, Conceptualization, Supervision, Funding acquisition, Investigation, Methodology, Project administration, Writing - review and editing, Writing - original draft

## Author ORCIDs
Katelyn Gostic (iD) https://orcid.org/0000-0002-9369-6371
Adam J Kucharski (iD) http://orcid.org/0000-0001-8814-9421
James O Lloyd-Smith (iD) https://orcid.org/0000-0001-7941-502X

## Decision letter and Author response
Decision letter https://doi.org/10.7554/eLife.55570.sa1
Author response https://doi.org/10.7554/eLife.55570.sa2

## Additional files

### Supplementary files
• Transparent reporting form

### Data availability
There are no data inputs into our model. All parameter input values are specified in Table 1, or in the manuscript text. We provide a link to the github repository containing all code necessary to run the analyses and generate figures (https://github.com/kgostic/traveller_screening/releases/tag/v2.1, copy archived at https://github.com/elifesciences-publications/traveller_screening).

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
