## [Decision Letter]

**Acceptance summary:**

This article revisits the model described by the same authors (Gostic et al., 2015) with updated parameters for an effective global response against the spread of COVID-19. The findings are based on a previously validated model and are epidemiologically plausible. They alert us to the fact that entry screening will miss the majority of infected travelers, a point that should help inform health surveillance in countries that serve as destination for travellers from China and other settings affected by the epidemic.

(Editorial Note: Following expedited peer review, the authors revised their paper by taking into account the science that had emerged on this topic up until February 20, 2020.)

**Decision letter after peer review:**

Thank you for submitting your article "Estimated effectiveness of traveller screening to prevent international spread of 2019 novel coronavirus (2019-nCoV)" for consideration by *eLife*. Your article has been reviewed by two peer reviewers, and the evaluation has been overseen by Eduardo Franco (Reviewing Editor) and Neil Ferguson (Senior Editor). The following individual involved in review of your submission has agreed to reveal their identity: James M McCaw. The other reviewer remains anonymous.

Your submission is a Research Advance; this format is for substantial developments that directly build upon a previous *eLife* paper. Our conclusion is that it does expand on your previous work, perhaps not to the extent that we usually like to see in such papers. However, in light of the importance of the topic vis-vis the ongoing COVID-19 epidemic, we intend to retain your paper for publication in *eLife* if you are able to revise it as per the terms below.

As is customary in *eLife*, the reviewers have discussed their critiques with one another. What follows below is my lightly edited compilation of the essential and ancillary points provided by reviewers in their critiques and in their interaction post-review. Some of the reviewers' comments may seem to be simple queries or challenges that do not prompt revisions to the text. Please keep in mind, however, that readers may have the same perspective as the reviewers. Therefore, it is essential that you attempt to amend or expand the text to clarify the narrative accordingly.

Summary:

This article revisits the model described by authors (Gostic et al., 2015) with updated parameters for COVID-19. The findings are based on a previously validated model and are epidemiologically plausible. They alert us to the fact that entry screening will miss the majority of infected travelers, a point that should help inform health surveillance in countries that serve as destination for travellers from China and other settings affected by the epidemic.

Title:

Please update title and text to reflect new WHO-approved terminology.

Essential revisions:

Parameters in Table 1 should be revisited with the latest literature, while acknowledging that information on COVID epidemiology continues to be updated from week to week. This would not change the overall conclusions of this study but would make it more up-to-the-minute with the rapidly changing epidemiology of this infection.

All choices are reasonable given current uncertainty of the clinical spectrum and epidemiological characteristics. 50% subclinical could do with some more explanation/justification. There have been some very recent estimates (from Trevor Bedford and from Imperial) of the possible total infection numbers. Likely, missing cases are symptomatic, perhaps mild. That may provide a slightly different and more data-informed bound to consider for this parameter. I note the Discussion comment in paragraph two so ultimately don't disagree with using 50% as one of the sampled values.

The breakdown of results as individual-level and population-level is clear (Figure 2 and Figure 3). I agree with this approach, and a note on why it is important to consider both levels, is made in the Introduction. A key requirement for the translation of work like this into policy/decision making is to clearly describe how an individual analysis misses key aspects of the problem, and cannot be used solely for decision making. The Introduction should cover that, providing a road-map for the paper.

Comment: Uncertainty representation in Figure 3 is excellent with (A) and (B).

Query: Given an LHS has been performed, can the authors use a PRCC to say something about what the primary drivers of the variability is (other than the screen-type and sub-clinical facets of course)?

---

## [Author Response]

Thank you for constructive reviews of our work. We respond to specific reviewer comments below. Here we note a few additional, voluntary additions to the revised manuscript, which we incorporated to address points relevant to the changing context of the COVID-19 epidemic, and to improve accessibility for a broad audience.

1) Our initial submission only considered screening effectiveness in a growing epidemic. Here, we add supplementary figures to consider outcomes in an epidemic that is no longer growing.

2) We added Figure panel 3C, which considers the probability of detecting the first *n* infected travellers that pass through screening. This new analysis addresses the possibility that screening could be used to delay, rather than to prevent case importations.

3) For ease of interpretation, we updated the way we report parameterization of the γ-distributed incubation period. The main text now reports the mean and standard deviation of this distribution rather than the shape and scale parameter. (The conversion should be straightforward for technical readers).

4) We have substantially revised the Introduction and Discussion to provide a more comprehensive view of the developing epidemic scenario, and to incorporate up-to-date information. In particular, we have reframed the title and Introduction to address the fact that symptom screening has become a ubiquitous tool for preventing not only international spread of COVID-19, but also local spread in affected cities and quarantined populations.

5) In the Discussion, we cite new data on the fraction of imported cases that had symptom onset after travel, to corroborate a key prediction of our model.

Title:Please update title and text to reflect new WHO-approved terminology.

Following official naming conventions adopted while our initial submission was in review, we now refer to the virus as SARS-CoV-2, and to the disease as COVID-19 throughout the manuscript.

Essential revisions:Parameters in Table 1 should be revisited with the latest literature, while acknowledging that information on COVID epidemiology continues to be updated from week to week. This would not change the overall conclusions of this study but would make it more up-to-the-minute with the rapidly changing epidemiology of this infection.

Of course. We added or updated eleven references in Table 1, which collectively provide new estimates for R 0, the incubation period mean and distribution, the upper-bound estimate of the subclinical case fraction, and the sensitivity of infrared thermal scanners.

We made the following quantitative adjustments to the values input into the model:

· Mean incubation period range: now 4.5-6.5 days, previously 4-7 days.

· Incubation period distribution: parameters updated to reflect new estimates.

· R 0 plausible range: now 1.5-4.0, previously 2.0-4.0.

Please note: at the time of resubmission, new information about COVID-19 is still emerging rapidly. Many of the estimates listed in Table 1 are still drawn from preprints, and may change as these studies are revised in peer review. However, we were reassured to see that new estimates added in this round of revision were broadly consistent with the original set compiled for the initial submission, and that only minor quantitative changes to model inputs were needed.

In addition to the changes noted above, we have substantially revised the Materials and methods, in which we provide additional details to justify the plausible parameter ranges considered by our model.

All choices are reasonable given current uncertainty of the clinical spectrum and epidemiological characteristics. 50% subclinical could do with some more explanation/justification. There have been some very recent estimates (from Trevor Bedford and from Imperial) of the possible total infection numbers. Likely, missing cases are symptomatic, perhaps mild. That may provide a slightly different and more data-informed bound to consider for this parameter. I note the Discussion comment in paragraph two so ultimately don't disagree with using 50% as one of the sampled values.

We carefully reviewed the available data, and ultimately decided that our original choices (5%, 25% and 50% subclinical cases) were consistent with data-driven estimates of the upper and lower bound. We have now revised the Materials and methods to more clearly justify our reasoning (subsection “Fraction of subclinical cases”).

We also now frame these choices explicitly as plausible upper-bound, lower-bound and middle case scenarios in the Results: “Additionally, we considered three possible fractions of subclinical cases: 50% represents a worst-case upper bound, 5% represents a best-case lower bound, and 25% represents a plausible middle case (Table 1, Materials and methods).”

Finally, we provide a more precise definition of “subclinical” in the context of our analysis: “We define subclinical cases as those too mild to show symptoms detectable in screening (fever or cough) after passing through the incubation period (i.e. once any symptoms have manifested). “

As explained in the revised text, our lower-bound (5%) is based on the fraction of clinically attended (i.e. relatively severe) cases that show fever or cough. Our upper-bound estimate is based on the fraction of cases asymptomatic on diagnosis, given active monitoring (on repatriation flights or cruise ships).

We considered the suggestion that estimates of the total case count could be compared to the official case count, but ultimately we did not feel confident in the estimates obtained using this approach. One challenge is that large numbers of cases are almost certainly going unreported not because they are mild, but because public health infrastructure has been overwhelmed in China. Further, official case counts are plagued by reporting delays of uncertain length, and official case definitions have changed since the start of the outbreak.

The breakdown of results as individual-level and population-level is clear (Figure 2 and Figure 3). I agree with this approach, and a note on why it is important to consider both levels, is made in the Introduction. A key requirement for the translation of work like this into policy/decision making is to clearly describe how an individual analysis misses key aspects of the problem, and cannot be used solely for decision making. The Introduction should cover that, providing a road-map for the paper.

We have modified the Introduction to explain the distinction between individual and population-level outcomes and we agree this helps frame the paper more clearly.

Comment: Uncertainty representation in Figure 3 is excellent with (A) and (B).

Thank you.

Query: Given an LHS has been performed, can the authors use a PRCC to say something about what the primary drivers of the variability is (other than the screen-type and sub-clinical facets of course)?

Thanks for this suggestion – we agree this analysis naturally follows on the results in the initial submission. As described in the revised Materials and methods and Results, we have added the requested analysis (Figure 4 shows the PRCC analysis for the middle-case assumption that 25% of cases are subclinical in a growing epidemic. Figure 4—figure supplement 1 shows all subclinical fractions in a growing epidemic, and Figure 4—figure supplement 2 shows all subclinical fractions in a stable epidemic).